# Clinical Outcomes after Additional Dynamic Renal^®^ Stent Implantation for Stent Recoil in Ostial Coronary Lesions

**DOI:** 10.3390/jcm9123964

**Published:** 2020-12-07

**Authors:** Bachir Abdulrahman, Kambis Mashayekhi, Péter Tajti, Miroslaw Ferenc, Christian Marc Valina, Willibald Hochholzer, Franz-Josef Neumann, Thomas Georg Nührenberg

**Affiliations:** Department for Cardiology and Angiology II, University Heart Center Freiburg-Bad Krozingen, 79189 Bad Krozingen, Germany; dr-abdulrahman@hotmail.com (B.A.); kambis.mashayekhi@universitaets-herzzentrum.de (K.M.); ptajti@gmail.com (P.T.); miroslaw.ferenc@universitaets-herzzentrum.de (M.F.); christian.valina@universitaets-herzzentrum.de (C.M.V.); willibald.hochholzer@universitaets-herzzentrum.de (W.H.); franz-josef.neumann@universitaets-herzzentrum.de (F.-J.N.)

**Keywords:** ostial coronary lesions, stent recoil, Dynamic Renal, drug-eluting stents

## Abstract

Background: Interventional treatment of aorto-ostial coronary stenoses is limited by stent recoil and suboptimal angiographic results, leading to restenosis and frequent re-interventions. As a potential bail-out strategy for stent recoil, implantation of an additional stent to increase radial force has been reported. Thus, we sought to investigate clinical outcomes after additional implantation of a Dynamic Renal^®^ stent (DRS), a non-coronary; bare-metal stent with very high radial force, in aorto-ostial coronary stenoses. Methods: Patients treated by implantation of DRSs for stent recoil in the ostial right coronary artery or the left main stem were identified from the hospital database. Baseline clinical and procedural characteristics were compared to patients who underwent re-intervention for in-stent-restenosis in similar segments by either implantation of conventional drug-eluting stents (DES) or paclitaxel-coated balloons (PCB). Clinical follow-ups were performed up to three years following re-intervention with the assessment of death, target lesion reintervention (TLR), and major adverse cardiac events (MACE) as a combination death, myocardial infarction and target vessel revascularization. Kaplan–Meier analyses were performed for event-free survival between the three groups. Results: Between 05/2013 and 07/2019, 28 patients underwent DRS implantation of aorto-ostial coronary lesions. In comparison with 49 patients with DES implantation and 29 patients undergoing PCB treatment, no relevant differences in baseline parameters were identified. Median follow-up was 714 days, with an available follow-up of >1 year after intervention in 82.1% of patients. In the entire study cohort at two years after re-intervention, the TLR rate was 16% (17 patients), the MACE rate 37% (39 patients), and all-cause mortality 9% (10 patients), with no significant differences between the three groups. Conclusions: DRS implantation for treating stent recoil of aorto-ostial coronary lesions resulted in a high rate of TLR, and was associated with similar risk for death and MACE compared to treatment of in-stent-restenosis with DES or PCB. Randomized, larger comparisons of contemporary DES in patients exclusively presenting with stent recoil are necessary to further define the efficacy and safety of this approach.

## 1. Introduction

Aorto-ostial coronary lesions such as stenosis of the ostial right coronary artery or the ostial left main stem have been recognized as challenging targets for percutaneous coronary intervention (PCI) due to an increased risk of restenosis which can be caused by stent recoil [1]. Stent recoil occurs when the inward pressure of fibrotic and calcified tissue at the coronary ostium exceeds the radial force of the implanted stent [2]. Despite implanting drug eluting stents (DES), restenosis remains a relevant clinical problem with high target lesion revascularization (TLR) rates one year after implantation of first-generation DES (12–28%) in right coronary ostial lesions [3,4]. Using newer generation DES, two registries reported TLR rates of around 12% in such lesions [5,6] which remains 3–4 times higher than in non-ostial lesions [7]. A recent analysis from an international multicenter registry showed that the use of newer-generation DES was associated with lower target lesion failure (TLF) rates whereas stent under-expansion was the strongest predictor for TLF, increasing the risk by 10-fold [8]. While the use of cutting balloons is still associated with a high rate of restenosis (16%) [9], PCI guidance by intravascular ultrasound (IVUS) and simultaneously implanting a stent with relatively high radial force showed the lowest TLR rates (5.3%) to date [10]. Compared to PCI of the right coronary ostium, PCI of the left main coronary ostium is associated with 10-fold lower risk of TLR [11]. However, if stent recoil occurred, implantation of a second stent to increase radial force has been reported feasible and safe in case reports [12,13] and small case series [14]. It appears conceivable that, in case of recoil after stent implantation of aorto-ostial lesions, the implantation of a second stent with high radial force may improve the angiographic result as well as the clinical outcome. The Dynamic Renal Stent^®^ (DRS, Biotronik, Berlin, Germany) is a bare metal stent with high radial force due to a cobalt-chromium alloy and struts of 120 µm thickness, thus thicker than struts of contemporary Resolute Onyx™ (81 µm for diameters ≤ 4.0 mm and 91 µm for diameters ≥ 4.5 mm) or Synergy Megatron™ (89 µm) DES. It was developed for the use in renal arteries, known for frequent recoil after angioplasty. 

In our center, the DRS has been used in selected cases with stent recoil or large diameter coronary arteries as compassionate use. The present study reports the clinical outcomes of patients that underwent implantation of a DRS as a second stent strategy in aorto-ostial lesions. 

## 2. Experimental Section

### 2.1. Study Population

This single-center, retrospective observational study sought to report clinical outcomes after treatment of aorto-coronary ostial stenoses with additional implantation of a dedicated recoil-resistant bare-metal stent (Biotronik Dynamic Renal^®^) upon recoil of a contemporary drug-eluting stent. Patients who underwent any DRS implantation were retrieved from the electronic health record database. Patients who were treated with DRS in lesions other than aorto-ostial lesions, i.e. the ostial right coronary artery or the left main stem, were excluded. Lesions were considered as ostial if they arised within 3 mm of the vessel origin. All angiograms and patient records were reviewed to verify that DRS implantation was due to observed stent recoil. Stent recoil was defined as remaining radiographic stent underexpansion after adequate expansion of a non-compliant balloon. Stent recoil before DRS implantation was visually graded into mild (≤25% of vessel diameter), moderate (>25% and ≤50% of vessel diameter) or severe (>50% of vessel diameter). After DRS implantation, recoil was re-assessed and graded as resolved, improved and unchanged. Patients forming comparator groups were equally identified from the electronic health record. To this end, patients that underwent repeated intervention with a second DES or with a paclitaxel-coated balloon (PCB) for relevant in-stent-restenosis in the left main stem or in the ostial right coronary artery were identified. Patients who experienced stent thrombosis or who were treated with plain balloon angioplasty were excluded. 

### 2.2. Collection of Data

For all patients, procedural characteristics of the repeat intervention, and clinical characteristics such as the presence of diabetes mellitus, hyperlipidaemia, hypertension, and smoking status as well as laboratory characteristics were also retrieved from the hospital database. Follow-up of patients undergoing PCI is routinely performed 30 days, 1 year, and 3 years after PCI and documented in the hospital database. 

### 2.3. Definition of Outcomes

Target lesion revascularization (TLR) within two years after the index procedure was defined as primary efficacy outcome. As secondary safety outcomes, the occurrence of all-cause death as well as or major adverse cardiac events (MACE), defined as a composite of all-cause death, myocardial infarction and target vessel revascularization (TVR) at 2 years were investigated. 

### 2.4. Statistical Analysis 

Discrete variables are reported as counts (percentages) and continuous variables as median with interquartile range. For discrete variables, we tested differences between groups with the χ2-test or Fisher’s exact test when expected cell sizes were less than 5. To compare continuous variables, the Kruskal–Wallis test was used. The incidence of clinical endpoints between the three treatment groups was compared by log-rank testing. Cumulative event rates were calculated and graphically described according to the Kaplan–Meier method. All tests were two-sided and results were regarded as statistically significant at an α-level of 5%. We used IBM SPSS statistics, version 23.0 (IBM corporation, Armonk, NY, USA) for all statistical analyses.

### 2.5. Ethics Statement

The study was approved by the ethics committee of the Albert-Ludwigs-Universität Freiburg, Germany, (number: EK-Freiburg 238/19) and is in accordance with the ethical guidelines of the 1975 Declaration of Helsinki, as revised in 1983.

## 3. Results

### 3.1. Characteristics of the Study Population

Between 05/2013 and 07/2019, 28 patients underwent DRS implantation due to recoil of conventional stents observed in aorto-coronary lesions. As a comparator group, 78 patients were identified to have undergone repeat intervention of aorto-ostial coronary lesions between 01/2013 and 12/2015. In this group, 49 patients were treated with a second-generation drug-eluting stent and 29 patients underwent balloon dilatation with subsequent paclitaxel-eluting balloon treatment (Figure 1). 

The median age of the patients treated with DRS was 75 years, while patients treated with DES or PCB tended to be younger with a median age of 71 years (*p*-value 0.09) (Table 1). In the DES and PCB groups, there was a trend to a higher rate of previous myocardial infarction as compared to the DRS group (*p*-value 0.08). Regarding cholesterol levels, there were differing trends for total cholesterol (*p*-value 0.06) and low density lipoprotein-cholesterol (LDL-cholesterol) (*p*-value 0.07), with the lowest level of total cholesterol in the DRS group and the lowest LDL-cholesterol in the PCB group. For all other variables, no significant differences or statistical trends were observed between the three groups. 

Regarding procedural characteristics, all patients in the DRS group had severe angiographic coronary artery calcification. Assessment of stent recoil in the DRS group is depicted in Table 2. Of note, recoil was improved or resolved after DRS implantation in 21 of 28 patients but remained unchanged in 7 patients. Procedural parameters pertaining to the reintervention are shown in Table 3 whereas parameters of the preceding stent implantation are depicted in Table 4. In our center, in-stent pre- and post-dilatations are routinely performed with non-compliant balloons.

### 3.2. Clinical Outcomes according to Treatment Strategy

During a median follow up of 717 (375–1114) days, TLR as primary outcome occurred in 4 out of 28 patients in the DRS group (14%), in 8 of 49 patients (16%) with DES implantation and in 6 of 29 patients (20%) with PCB treatment. In Kaplan Meier analyses (Figure 2A), these differences were statistically not significant (*p* log-rank test 0.84). The risk for MACE as a safety outcome, defined as a composite of all-cause death, myocardial infarction and TVR), was statistically not different between the groups (*p* log-rank test 0.90), occurring at a rate of 32% (9 of 28 patients) in the DRS group, 41% (20 of 49 patients) in the DES group and 38% (11 of 29 patients) in the PCB group (Figure 2B). Regarding all-cause death, three patients (11%) in the DRS group died, compared to five patients (10%) in the DES group and two patients (7%) in the PCB group (Figure 2C, *p* log-rank test 0.81).

## 4. Discussion

Long-term outcomes of PCI in aorto-ostial coronary lesions with DES is less favorable compared to the results of PCI in non-ostial lesions [5,6]. Several approaches, such as intravascular imaging and implantation of a stent with high radial strength [10], may improve the long-term success after PCI in this setting. Double Stenting for stenting recoil has been reported in case reports and small case series [12,13,14]. 

This study reports for the first time the use of a bare-metal stent with very high radial strength, the Dynamic Renal stent, as a second stent for lesions with observed stent recoil. Compared to previous case series of double-stenting in aorto-ostial coronary lesions, our study comprises more patients and compares the results of DRS implantation to comparator groups treated with DES and PCB. 

Overall, we did not observe relevant differences between the three treatment strategies. Nevertheless, these results should not be used at first sight to argue against the implantation of a DRS in cases of stent recoil in aorto-ostial coronary lesions. Conversely, it must be taken into account that the patients in the comparator groups were treated for in-stent restenosis that occurred after initial, successful implantation of a DES whereas DRS implantation was carried out as a bail-out strategy in cases of visible stent recoil. 

It is also noteworthy that DRS implantation did not uniformly resolve the observed stent recoil. In our opinion, optimal lesion preparation by debulking of calcifications through rotational atherectomy before placement of the first stent must thus be emphasized. In our study, the PCB group had a more aggressive lesion preparation compared to the DES group. This might serve as an explanation for the comparable outcome of PCB treatment not adding radial force but further supporting the importance of lesion preparation. Also, ultrasound guidance for stent placement has been associated with better outcomes [10,15]. This effect on outcomes may be explained by the ability of IVUS to reduce geographical miss, whereas two-dimensional angiography frequently does not result in complete lesion coverage [16]. Therefore, routine use of IVUS in aorto-ostial lesions should be considered. Furthermore, stents with high radial strength such as the Resolute Onyx™ stent (Medtronic, Dublin, Ireland) or the Synergy Megatron™ (Boston Scientific, Marlborough, Massachusetts, USA) stent should be preferred over stents with thin struts and less radial strength in order to prevent stent recoil. If, however, stent recoil is observed despite the outlined procedural measures, we show that double stenting with DRS can be an option to reduce or resolve stent recoil. Novel treatment options such as intravascular lithotripsy [17] showed promising results [18] which need to be confirmed in randomized trials and in the context of aorto-ostial coronary lesions. When restenosis or stent recoil is noted after previous PCI of right coronary lesions, bypass surgery should be the preferred treatment strategy if concomitant left coronary revascularisation is necessary.

Certainly, our study has some limitations. First, it is a non-randomized retrospective analysis and selection bias could have occurred. However, patients with DRS as well as comparator groups were retrieved electronically by predefined criteria independent of individual patient characteristics. Second, the overall number of patients treated for stent recoil with DRS is still small, yet, this is to date the largest series of double stenting for stent recoil. Third, due to the observational, retrospective character of the study, no routine assessment of stent recoil by intravascular ultrasound or enhanced stent visualization technology was performed. Fourth, the comparator groups were not primarily groups with stent recoil but indeed patients with relevant in-stent-restenosis. In this context, it must be noted that stent recoil is usually caused by heavy calcifications whereas in-stent-restenosis is due to exaggerated extracellular matrix deposition. Nevertheless, both conditions are associated with high TLR rates and underline the need for more effective treatment options for aorto-ostial coronary lesions.

In conclusion, the present data demonstrate that in cases of stent recoil in aorto-ostial coronary lesions, the implantation of a second, bare-metal stent with high radial force remains associated with high TLR and MACE rates at two years of follow-up. Further studies are needed to evaluate whether the outcome after PCI of aorto-ostial coronary lesions can be reconciled to outcomes of PCI for non-ostial lesions. These studies should specifically address which combination or selection of ultrasound guidance, lesion modification and stents with high radial force allows to achieve high success rates in PCI in aorto-ostial coronary lesions. 

## Figures and Tables

**Figure 1 jcm-09-03964-f001:**
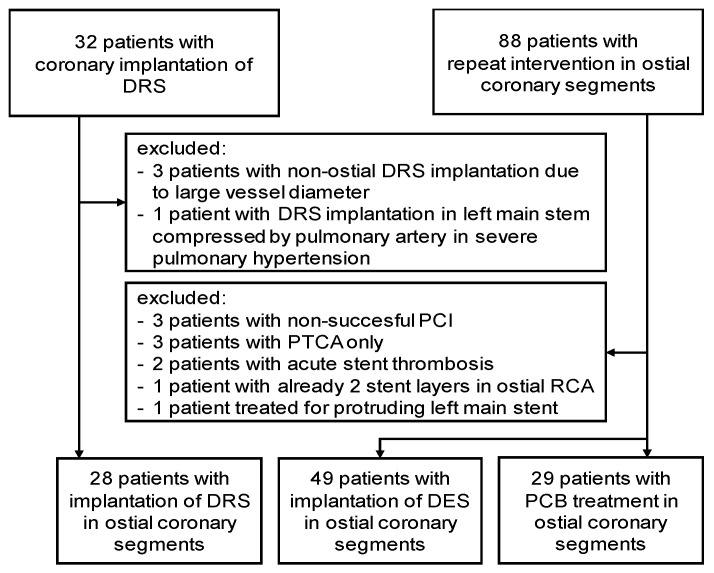
Study flow chart. DRS: Dynamic Renal stent, DES: drug-eluting stent, PCB: paclitaxel-coated balloon, RCA: right coronary artery. PCI: percutaneous coronary intervention; PTCA: percutaneous transluminal coronary angioplasty.

**Figure 2 jcm-09-03964-f002:**
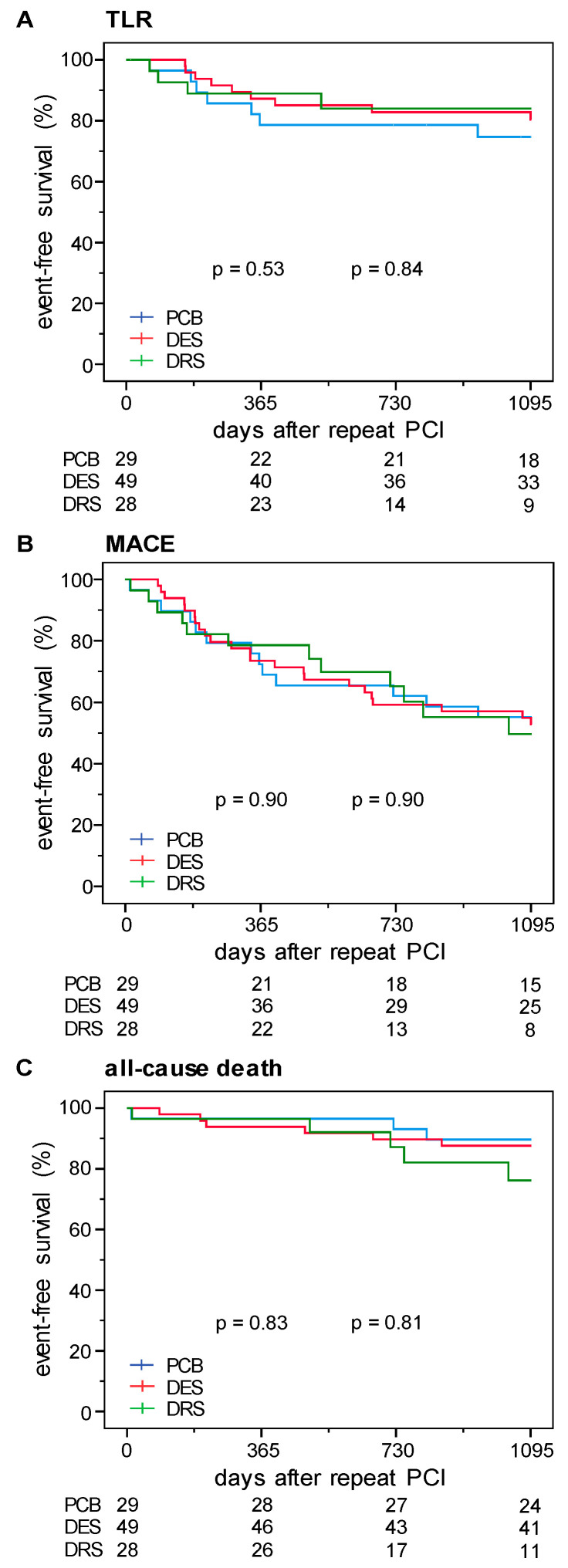
Kaplan–Meier estimates for cumulative event rates of (**A**) target lesion revascularization (TLR), (**B**) major adverse cardiac events (MACE), and (**C**) all-cause death in patients treated with paclitaxel-coated balloon (PCB, blue), drug-eluting stent (DES, red), or Dynamic Renal stent (DRS, green) treatment. *p*-value by log-rank test; PCI: percutaneous coronary intervention.

**Table 1 jcm-09-03964-t001:** Patient Characteristics.

	DRS (*n* = 28)	DES (*n* = 49)	PCB (*n* = 29)	*p*-Value
Median Follow-up (days)	992 (514–1108)	1312 (1110–2137)	1550 (111–2140)	<0.01
Follow-up at 2 years complete	21 (75%)	48 (98%)	29 (100%)	<0.01
Follow-up at 1 year complete	27 (96%)	49 (100%)	29 (100%)	0.25
Right coronary ostial lesion	22 (78%)	39 (80%)	18 (62%)	0.20
Age (years)	75.5 (70.2–80.0)	71.0 (60.5–77.0)	71.0 (63.5–78.0)	0.09
Female sex	8 (28.6%)	9 (18.4%)	7 (24.1%)	0.57
BMI (kg/m²)	27.8 (25.7–30.9)	27.3 (24.8–30.8)	26.8 (23.8–30.0)	0.44
**Risk factors**				
Arterial hypertension	26 (92.9%)	43 (87.8%)	27 (93.1%)	0.65
Diabetes mellitus	8 (28.6%)	16 (32.7%)	13 (44.8%)	0.39
Family history of CVD	6 (21.4%)	24 (49%)	11 (37.9%)	0.05
Current smoking	2 (7.1%)	7 (14.3%)	3 (10.3%)	0.62
**Cardiac history**				
Acute coronary syndrome	5 (17.9%)	5 (10.2%)	9 (31%)	0.06
Coronary artery bypass grafting	7 (25%)	8 (16.3%)	10 (34.5%)	0.18
Previous myocardial infarction	6 (21.4%)	15 (30.6%)	14 (48.3%)	0.08
Reduced left ventricular function	9 (32.1%)	13 (26.5%)	8 (27.6%)	0.86
**Medication**				
Length of P2Y12 inhibition	12 (6.0–12)	6 (6.0–6.0)	6 (3.75–11.5)	<0.01
Insulin	3 (10.7%)	8 (16.3%)	6 (20.7%)	0.58
Oral antidiabetics	6 (21.4%)	12 (24.5%)	9 (31%)	0.69
**Laboratory parameters**				
eGFR (mLmin)	68.6 (55.2–78.0)	72.6 (55.1–77.7)	72.8 (56.9–85.5)	0.56
Hemoglobin (g/dL)	13.3 (12.0–14.3)	14.3 (13.1–15.3)	13.9 (12.0–14.5)	0.12
Total cholesterol (mg/dL)	150 (134–168)	169 (142–204)	156 (141–188)	0.06
LDL cholesterol (mg/dL)	93 (65–110)	100 (82–133)	89 (72–107)	0.07
HDL cholesterol (mg/dL)	50.0 (42.0–65.7)	49.0 (40.0–60.5)	50.0 (43.5–64.0)	0.52
C-reactive protein (mg/dL)	0.3 (0.1–0.8)	0.3 (0.1–0.5)	0.2 (0.1–0.4)	0.69

DRS: Dynamic Renal stent; DES: drug-eluting stent; PCB: paclitaxel-coated balloon; *p*-value by log-rank test; BMI: body mass index; CVD: cardiovascular disease; eGFR: estimated glomerular filtration rate; LDL: low density lipoprotein; HDL: high density lipoprotein; P2Y12: adenosine diphosphate receptor type 12.

**Table 2 jcm-09-03964-t002:** Procedural Characteristics of Patients with Stent Recoil.

	DRS (*n* = 28)
Severity of recoil (mild/moderate/severe)	15/12/1
Recoil after DRS (unchanged/improved/resolved)	7/11/10
Procedural optimization and auxiliary devices	
Guide catheter extension	3
Cutting-Balloon	5
Ultra-high pressure balloon (OPN)	7
Intravascular ultrasound	3
Rotational atherectomy	2

DRS: Dynamic Renal stent.

**Table 3 jcm-09-03964-t003:** Procedural Characteristics of the Re-intervention.

	DRS (*n* = 28)	DES (*n* = 49)	PCB (*n* = 29)	*p*-Value
Predilatation performed	28 (100%)	39 (78%)	28 (97%)	0.007
Diameter (mm)	4.0 (3.0–4.0)	3.0 (2.5–3.5)	3.25 (3.0–3.5)	0.001
Maximal pressure (bar)	22 (18.5–26)	17 (16–22)	20 (13.25–23.5)	0.014
Stent/PCB diameter (mm)	5.0	3.5 (3.5–4.0)	3.5 (3.0–4.0)	<0.001
Stent/PCB length (mm)	12	16 (12–20)	15 (15–20)	<0.001
Maximal pressure (bar)	16 (12.5–20)	18 (14–20)	14 (12–16)	<0.001
Stent type				
Promus Element/Premier		12 (25%)		
Resolute Integrity		12 (25%)		
Xience Pro		9 (18%)		
Orsiro		5 (10%)		
Synergy II		4 (8%)		
Other		7 (14%)		
Postdilatation Performed	24 (86%)	28 (57%)	0 (0%)	<0.001
Diameter (mm)	4.25 (4.0–5.0)	4.0 (3.5–4.0)	n.a.	0.003
Maximal pressure (bar)	21 (20–40)	20 (20–24)	n.a.	0.069

DRS: Dynamic Renal stent; DES: drug-eluting stent; PCB: paclitaxel-coated balloon; *p*-value by log-rank test; n.a.: not applicable.

**Table 4 jcm-09-03964-t004:** Procedural Characteristics of the Initial Intervention.

	DRS (*n* = 28)	DES (*n* = 49)	PCB (*n* = 29)	*p*-Value
Stent diameter (mm)	3.5 (3.5–4.0)	3.5 (3.25–4.0)	3.5 (3.0–3.5)	<0.001
Stent length (mm)	16 (15–24)	18 (15–25)	26 (18–33)	0.015
Maximal pressure (bar)	18 (18–20)	16 (14–18)	16 (16–17)	<0.001
Stent type				
Promus Element/Premier	10 (36%)	6 (12%)	9 (31%)	
Resolute Integrity	2 (7%)	10 (20%)	8 (28%)	
Resolute Onyx	7 (25%)			
Xience Pro	4 (14%)	5 (10%)	2 (7%)	
Orsiro	0	1 (2%)	1 (3%)	
Synergy II	2 (7%)	0	0	
Other	3 (11%)	27 (55%)	9 (31%)	
Postdilatation Performed	28 (100%)	20 (41%)	13 (45%)	<0.001
Diameter (mm)	4.0 (3.5–4.5)	3.5 (3.5–4.0)	3.5 (3.5–4.5)	0.004
Maximal pressure (bar)	24 (20–40)	20 (16.5–22)	20 (16.5–22)	0.010

DRS: Dynamic Renal stent; DES: drug-eluting stent; PCB: paclitaxel-coated balloon; *p*-value by log-rank test.

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
