# Peer review of "Clinical Outcomes after Additional Dynamic Renal® Stent Implantation for Stent Recoil in Ostial Coronary Lesions"

_jcm, 2020, doi:10.3390/jcm9123964_

Round 1

Reviewer 1 Report

PCI of aorto-ostial lesions, especially those located in the right coronary artery (RCA) remains challenging with high risk for procedural failure or suboptimal angiographic results, due to  technical difficulties to cover the ostium with stent, risk for stent recoil and the presence of moderate/severe calcifications that increase the risk for restenosis and adverse events. In the present study, Abdulrahman et al, in this retrospective study evaluated the efficacy and safety of DRS-BMS for the treatment of aorto-ostial lesions. The main outcome was the risk for TLR at 2 years follow up.

However, there are many major issues with this report:

  1. The study population is too small to make any conclusions about the efficacy and safety of DRS stent.
  2. The authors stated that ”angiograms and patient records were reviewed to verify that DRS implantation was due to observed stent recoil”. However, they did not provided any definition about ”stent recoil”, the severity of stent underexpansion or the underlying mechanism
  3. Appropriate lesion preparation, with predilatation, non-compliant balloon, cutting or scoring balloon and rotational atherectomy or lithotripsy, the use of IVUS and postdilatation with OPN balloon in selected cases are crucial in order to improve outcomes. In selected cases with recoil, implantation of a second contemporary DES may be considered.
  4. The authors have not provided any data about lesion-characteristics, procedural characteristics or stent data (diameter, length, maximum pressure, post-dilatation etc). Furthermore, this study did not provide any data suggesting that DRS, should be superior or at least non-inferior to second DES implantation (that is available in all cath lab and approved for coronary vessels). Furthermore, In cases with severe acute recoil after stent implantation, implantation of a second contemporary stent is challenging because it is difficult to cross the ”area of recoil-underexpansion” with the second stent. I assume that it should be more difficult to implant a ”bulky” DRS stent.

  1. The study aim was “to evaluate whether treatment of aorto-coronary ostial stenoses with additional implantation of a dedicated recoil-resistant bare-metal stent (Biotronik Dynamic Renal®) upon recoil of a contemporary drug-eluting stent is equally effective and safe compared to alternative implantation of a second DES or treatment with a paclitaxel-coated balloon (PCB)

However, this statement is misleading because the control group consisted of DES-ISR lesions and not lesions with recoil, treated with an additional DES. To compare DRS with DES-ISR lesions is not appropriate, DES-ISR lesions represent a very high-risk group for a new event, regardless the treatment.

  1. DRS treated lesion were included between 2013 and 2019 but comparators between 2013 and 2015. Although the authors provided the median (IQR) for follow up in the total population, the median follow up in the control groups should be longer.

  1. There are major differences in baseline characteristics, but the small study size did not allow any adjustment

  1. “Outcomes were derived from the hospital database. Follow-up of patients undergoing PCI were routinely performed 30 days, 1 year and 3 years after PCI and documented in the hospital database.”

But how many patients were lost during follow up at 2 years? In the KM curves for TLR, in the DES group of the total population of 49 patients, 7 had TLR and 5 died at 2 years but only 16 remains at risk at 2 years that does not make any sense…..In case that a patient was lost during follow up or did not have 2 years follow up (patients included 2018-2019), how the authors did get data about restenosis? How can the authors in those cases rule out that some patients did not undergo a new angiography in other hospital?

  1. To show that DRS implantation is feasible the authors should provide data about successful implantation. In how many cases the operators intended to implant a DRS but failed?
  2. Given the very small sample size, Subgroup analysis per vessel is not appropriate. Furthermore, aorto-ostial lesion in LM vs RCA may be completely different lesions
  3. Clinically driven restenosis 14% is very high,  angiographic restenosis would be considerably higher
  4. In discussion: “ in patients not amenable to surgery or patients with single vessel disease of the right coronary artery, debulking of calcifications through rotational atherectomy may be considered” is incorrect. Rota cannot be used after unsuccessful stent implantation
  5. “In conclusion, the present data demonstrate that in cases of stent recoil in aorto-ostial coronary lesions, the implantation of a second, bare-metal stent with high radial force has been shown feasible and safe.” This study is not powered to support this conclusion

Reviewer 2 Report

This is an interesting retrospective observational study comparing the feasibility and long term follow-up of aorto-ostial in-stent restenosis (ISR) treatment by a renal bare metal stent with a high radial force vs DES or Paclitaxel coated balloon. However, the study is limited by its retrospective nature. It is not clear whether all cases of aorto-ostial restenosis have been included in the comparator group (78 for the very busy cathlab of Freiburg and Bad-Krozingen appears to be small in 3 years).

It was difficult to rate overall the manuscript, missing all tables and figures in the downloadable document.

Some discussion of the following remarks might contribute to the article:

  • Adding a well-structured definition of aorto-ostial lesions
  • Was there any geographical miss during the initial PCI? How often was IVUS used (that is well stated by the authors that it decreases TLR)? Was IVUS performed at the time of the ISR treatment? Were stent struts malapposed? Which stent types were used, initially and for ISR treatment? It would be interesting to review the radial force of these stents critically in view of recent publications.
  • Line 71 “in” is used twice in a row
  • Regarding the results of the study: MACE included all cause death and all cause mortality was repeated which brings some confusion. Could the authors elaborate?
  • Was there any cardiac death? Any stent thrombosis?
  • Time of DAPT should be included in the patients’ characteristics section
  • Regarding the characteristics of study population: add the number of RCA vs LM in the different groups if possible
  • How was stent recoil assessed? Was it by the use of IVUS? Did the primary stent have the proper diameter size?
  • Of note, stent thrombosis were an exclusion criteria, although it might well be secondary to an important ISR...and cases interesting to report
  • Recent advances in stent structure and scaffold design would permit nowadays to have a better radial force in aorto ostial and calcified lesions treated by DES (SCAAR and EVOLVE registries)
  • Finally, could the author comment on the interestingly good results obtained with drug coated balloons...that brings no additional radial force
  • We understand that there was no power calculation since it was a retrospective analysis, but could the author elaborate on the number of patients, and numbers of years to look at, in order to have groups large enough to demonstrate a difference in clinical outcome?

Round 2

Reviewer 1 Report

The authors have successfully addressed all my comments in the revised version of the manuscript. 

Althought DRS use in coronary vessels remains an off-label indication and a RCT, powered to evaluate the efficacy and safety of DRS vs second DES is not expected, DRS use may be a bailout treatment in highly selected cases. Therefore, the results of the present study may be intresting.